# Recent Advances in the Spintronic Application of Carbon-Based Nanomaterials

**DOI:** 10.3390/nano13030598

**Published:** 2023-02-02

**Authors:** Shweta Pawar, Hamootal Duadi, Dror Fixler

**Affiliations:** 1Faculty of Engineering and the Institute of Nanotechnology and Advanced Materials, Bar Ilan University, Ramat Gan 5290002, Israel; 2Bar-Ilan Institute of Nanotechnology & Advanced Materials (BINA), Bar Ilan University, Ramat Gan 5290002, Israel

**Keywords:** carbon-based spintronics, graphene, carbon nanotube, fullerenes, carbon nitride

## Abstract

The term “carbon-based spintronics” mostly refers to the spin applications in carbon materials such as graphene, fullerene, carbon nitride, and carbon nanotubes. Carbon-based spintronics and their devices have undergone extraordinary development recently. The causes of spin relaxation and the characteristics of spin transport in carbon materials, namely for graphene and carbon nanotubes, have been the subject of several theoretical and experimental studies. This article gives a summary of the present state of research and technological advancements for spintronic applications in carbon-based materials. We discuss the benefits and challenges of several spin-enabled, carbon-based applications. The advantages include the fact that they are significantly less volatile than charge-based electronics. The challenge is in being able to scale up to mass production.

## 1. Introduction

In the 1990s, the term spintronics (short for spin electronics) was created to designate technologies that make use of an electron’s quantum mechanical feature known as “spin”, which has only two values: spin-up and spin-down. To keep up with the ever-increasing data generation, our society has dramatically boosted data storage capacity. Memory devices have shrunk in size concurrently [1]. In part, this growth in data storage capacity at smaller dimensions makes it easier to carry and exchange online material such as music and photographs on mobile devices. This field of study and technology is frequently referred to as spintronics since a significant portion of this technological breakthrough has been made possible by controlling the spin degree of freedom of electrons in circuits. After the giant magnetoresistance (GMR) effect was discovered in the 1980s, spintronics research developed pace [2]. One of the newer technologies that will allow next-generation nanoelectronics devices to have more memory and processing capability while using less electricity is spintronics. Such devices make use of the degree of freedom that electrons and/or holes’ spins have, which can also interact with their orbital moments. The spin polarization in these devices is managed either by magnetic layers acting as spin-polarizers or spin-polarizer/analyzers or by spin–orbit coupling [3].

The carbon material research focuses on carbon nanotubes and graphene. Although carbon exists in a variety of allotropic forms, these two substances offer a low dimensionality and high crystalline purity, opening the door to several unusual quantum events [4]. Spin injection and transport in carbon materials, such as carbon nanotubes, graphene, fullerene, and organic materials, are referred to as “carbon-based spintronics”. They can display strong long-range magnetic ordering and have much applicability if magnetic moments are introduced. For application in computers, tablets, and handheld devices, spintronics is utilized in fast microchips, logic gates, transistors, and capacitors. Additionally, all-hydrocarbon products can drastically cut the demand for crucial materials used in the semiconductor sector (e.g., indium and gallium) [3,5]. It is necessary to comprehend the history of magnetic coupling and make improvements to this vital characteristic to create new and significantly improved spintronic devices. At the same time, traditional research into hybrids of pure carbon units and magnetic metal species, known as magnetically functionalized carbon nanostructures, continues to advance rapidly. When creating composites with tunable magnetic characteristics, carbon components are essential because they serve as templates for magnetic units. Various kinds of carbon nanomaterials are utilized to shield and stabilize encapsulated magnetic subunits by forming cages with high cohesion [6,7,8].

To properly understand carbon-based spintronics, it is crucial to assess the most recent research in the area. This review article has explored the evolution of carbon-based spintronics and has also discussed some current problems and anticipated future developments. The preparation of the materials needed for carbon-based spintronic is covered in the first section. The second section illustrates the operation of carbon-based spintronic device operations. In this section, a few potential carbon-based spintronic devices for logic and memory applications are introduced along with other uses. The last section discusses the topic of prospective and challenges in carbon-based spintronics for real-world applications.

## 2. Carbon-Based Spintronics

### 2.1. General

The different carbon prototypes used in the synthesis serve as the foundation for most carbon-based spintronic devices. The carbon materials utilized are divided into four distinct types: graphene-based, carbon nanotubes-based, carbon nitride-based, and fullerenes-based (Figure 1). These are representations of carbon’s nanoscopic allotropes in two, one, and zero dimensions.

### 2.2. Preparation

It is possible to think about graphene as a single sheet of graphite since it produces a perfectly planar layer of carbon atom. Thus, the arrangement of carbon atoms results in a hexagonally symmetric, monatomic, two-dimensional lattice. Currently, there are three main methods for creating graphene nanoribbons (GNRs): cutting graphene using lithography, polycyclic bottom-up synthesis, and unzipping carbon nanotubes (CNTs). Chemical vapor deposition (CVD) on metal surfaces, mechanical exfoliation, electrochemical exfoliation, chemical exfoliation of graphite powder, epitaxial growth on single crystal SiC, chemical coupling processes, and intercalation/sonication are the common techniques used to produce graphene nanosheets (GNs) [9]. Graphene quantum dots (GQDs) can be created using top-down or bottom-up methods. In the top-down strategy, different synthetic and fabrication techniques are used to create zero-dimensional GQDs from specific carbon source materials (carbon fibers, graphene oxide, and graphene nanoribbon) [10]. These include microwave-assisted hydrothermal, solvothermal, chemical oxidation, microfluidization, and electrochemical techniques. Using a bottom-up strategy, stepwise solution chemistry, pyrolysis and oxidation, and hydrothermal heating are used to combine smaller benzene derivative units into larger GQD entities. The single-step Wurtz reaction is used to create scalable carbon nanosheets (CNs) without magnetic impurities, which feature an amorphous structure with embedded crystalline graphene nanocrystals by Liu C. et al. [11]. The synthesis is depicted in Figure 2.

Arc discharge [12], chemical vapor deposition, and laser ablations are some of the techniques used to produce CNTs [13]. Most frequently, reactive nitrogen-rich, oxygen-free molecules with pre-bonded C-N core structures, such as triazine and heptazine derivatives, are utilized as precursors in the chemical production of gC3N4. Using conveniently accessible nitrogen-rich precursors such as cyanamide, urea, thiourea, melamine, and dicyandiamide, it can be produced with just one step of heat processing [14]. Initially, fullerenes were created by laser vaporizing carbon in an inert atmosphere, but this process only yielded trace amounts of fullerenes. Later on, using the arc heating of graphite and the laser irradiation of polyaromatic hydrocarbon (PAHs), substantial amounts of fullerene C60 were produced. The synthesis process for different carbon-based materials is illustrated in Figure 3.

## 3. Applications

### 3.1. Graphene-Based

This carbon-based substance, graphene, is fundamentally important in spintronics and the reason for this decision was due to its inherent magnetic properties. The dimensionally reduced graphene nanoribbons are introduced in this section along with a general description of graphene. The zigzag varieties of these species, which have been shown to display magnetic properties in their ground state, can be viewed as the basic elements of carbon nanostructure magnetism. With a 1.97 eV optical bandgap, carbon nano solenoid (CNS) exhibits strong red photoluminescence. According to the results of magnetic tests, the CNS exhibits complicated magnetic ordering behavior and a paramagnetism response at low temperatures. Overall, such a Π-conjugated CNS makes it possible to serve as the foundation for the creation of electrical and spintronic devices that comprise of CNS molecules [15]. Partially hydrogenated graphene (PHGr), produced by S. Yang and his research team, contains periodic hexagonal graphene nanoflakes (GNFs) with zigzag borders embedded within a layer of hydrogenated graphene [16]. They revealed that a single Au atom can catalyze H_2_ dissociation by selectively adhering to borders, lowering the barrier to graphene hydrogenation. The PHGr can therefore be used as a platform for single-atom catalysts and carbon-based spintronic devices. Additionally, when the size of the GNF fluctuates between 1.4 nm and 2.3 nm, it was discovered that the antiferromagnetic boundary states and semiconducting characteristics remain unaltered. The impact of magnetic coupling engineering on the spin transport characteristics of covalently coupled nanographene dimer-based molecular devices has been investigated by Jing Zeng and Xiaohui Deng. Their findings demonstrate that the magnetic coupling strength of the covalently connected, five-membered ring-decorated nanographene is a key factor in the superior spin transport capabilities [17]. The on-surface synthesis of covalently bound triangulene dimers and a proof-of-concept experimental analysis of magnetism are presented by Mishra S. et al. The selective synthesis of triangulene dimers, in which the triangulene units are either directly joined through their minority sublattice atoms or are separated by a 1,4-phenylene spacer, results from on-surface interactions of rationally chosen precursor molecules on Au(111) [18]. Bond-resolved scanning tunneling microscopy has been used to describe the chemical composition of the dimers. Collective singlet–triplet spin excitations are discovered in the dimers by scanning tunneling spectroscopy and inelastic electron tunneling spectroscopy experiments, proving effective intertriangulene magnetic coupling. Using 2,6-dibromo-1,5-diphenylnaphthalene as a precursor, Keerthi A. et al. describe the on-surface synthesis of an unheard-of cove-edge chiral GNR with a benzo-fused backbone on an Au(111) surface [19]. A scanning tunneling microscopy and spectroscopy indicate the initial precursor self-assembly and the development of the chiral GNRs following annealing, as well as a very tiny electronic bandgap of about 1.6 eV.

The “inverse Schottky” feature, high spin polarization, the reverse of spin polarization, and substantial magnetoresistance were among the remarkable spin transport properties discovered. These findings suggest that the covalently bonded, five-membered ring-decorated nanographene dimers can be engineered to function as overcurrent protection, information storage, and logic devices. The research supports a practical approach to producing high-performance molecular spintronic devices made of carbon. Different emergent spin-related quantum phenomena are strongly influenced by spin–orbit coupling (SOC). Additionally, light carbon atoms imply a small intrinsic SOC strength, which prevents their use in spintronics. Gu J. et al. develop a specific deformation vector with chiral curvature, to replicate the warping and twisting of space to increase the SOC effect [20]. This opens the door to the creation of numerous spintronic devices, such as spin current sources, spin field-effect transistors, spin galvanic meters, and inverse spin Hall effect-related spin current sources and transistors. The discovery by de Sousa, M.S.M. et al. that periodic vacancies can transform single-layer graphene into a nodal-line or nodal-loop semimetal opens the door to the creation of novel capabilities for spintronic or electrical devices based on graphene [21]. Additionally, the density of states at the Fermi level is increased in the vacancy-engineered graphene. This method is true even in the presence of high spin–orbit coupling, regardless of the specifics of the systems, and applies to a variety of crystalline structures. The isolated cyclo [18] carbon (C18), a recently synthesized carbon allotrope, has the potential to be the essential element of advanced electronic systems because of its superior physical characteristics. Based on the non-equilibrium Green’s function approach and density functional theory, Hou L. et al. studied a unique molecular spin-filtering device that was created by covalently sandwiching a C18 ring between two zigzag-edged graphene nanoribbons (ZGNRs) [22]. By sandwiching a metal-carbon chain between C18 and nonmagnetic ZGNRs (on/off ratio of ≈2737), it is possible to produce a potent bimolecular configuration switching effect. Correa J.D. et al. systematically investigate the electrical and magnetic properties of a class of Penta-graphene-related materials that are produced by swapping out the four-fold coordinated carbon atoms for other elements [23]. Pentagonal nanoribbons exhibit remarkable electrical and magnetic properties as a result of quantum confinement and edge effects. These systems can retain magnetic states with different electronic behaviors or spin-unpolarized states, depending on the particular pentagonal material and edge geometries. Liang Z.’s group uses density functional theory and non-equilibrium Green’s function to conduct a theoretical study on the electrical characteristics of double atomic carbon chains bridging graphene electrodes. Atomic carbon chains’ conductivity is significantly impacted by strain. But the intrinsic transport of double atomic carbon chains is mostly governed by the coupling effect between adjacent chains. The connected double atomic chains have strong antiparallel spin-filtering capabilities on two electrodes. In spintronic devices and carbon-based field-effect transistors, the connected double-atomic carbon chains have a wide range of possible applications. An all-carbon spintronic device made of a perylene molecule connected to two symmetrical ferromagnetic zigzag-edge graphene nanoribbon (ZGNR) electrodes via carbon atomic chains was the subject of an investigation by Han X. et al. [24]. The ZGNR electrodes magnetization could be changed by applying an external magnetic field. The setup displays spin filtering and the negative differential resistance (NDR) effect when the spin configuration is parallel. Antiparallel spin arrangement reveals bipolar spin filtering, spin rectifying, and a NDR effect. Additionally, there is a significant magnetoresistance difference between the parallel and antiparallel spin arrangements. By organizing the direction of the magnetic moments of carbon atoms at the edges, Prayitno T.B., Budi E. and Fahdiran R. have shown how first-principles calculations may be used to modify the band gap of the bilayer zigzag graphene nanoribbon [25]. The polar angle, as it is specified in spherical coordinates, was used to specify these directions. The band gap increases as the polar angle rises from the ferromagnetic configuration to the antiferromagnetic state. In addition, it has been demonstrated that the ferromagnetic configuration leads to the metallic system, whereas the others lead to the insulator. A straightforward plasma technique was used by Jeong et al. to create N-doped graphene, and the resulting material was shown to have a specific capacitance of 280 F/g, which is four times greater than that of the similar undoped pure graphene [26]. This is due to the fact that N-doping can improve the electrical conductivity of graphene and introduce charge-transferring sites through a doping-induced charge modulation, leading to an increase in the specific capacitance and an improvement in power density of 8105 W/kg and energy density of 48 Wh/kg. GO can also be hydrothermally reduced using compounds that contain nitrogen to create N-doped graphene.

#### 3.1.1. Doping or Embedding

The structure of nanoscale carbon materials can be improved and controlled through doping, which in turn affects their optical, electrical, dielectric, and magnetic properties. It was shown that the system’s spin characteristics can be dramatically affected by cobalt (Co) atoms chemisorbed on p-conjugated C atoms. Distinct Co atoms have different distributions of the spin densities of the various energy levels, creating pathways for effective spin-transfer operations. Both global spin-transfer activities between the Co atoms and reversible local spin-flip events on each Co atom are accomplished [27]. Zhang P. and colleagues showed that the magnetism in boron atoms (B2) doped seven-atom-wide armchair graphene nanoribbons (B2-7AGNRs) is provided by p-electrons, resulting from the imbalance of electrons in two spin channels because of boron dopants [28]. The significant observation of R. Langer and his team was that doping graphene with two separate transition metal atoms and creating transition metal dimers results in a vastly increased magnetic anisotropy energy (MAE) when compared to graphene doped with a single atom [29]. A new two-dimensional carbon allotrope called twin T-graphene that has three atomic layers of thickness was explored by Majidi R. et al. [30]. Twin graphene is created by substituting carbon dimers for one-third of the parallel aromatic bonds in AA-stacked bilayer graphene. It resembles a graphene bilayer in which two layers are connected organically [31]. In their study, density functional theory (DFT) calculations were used to examine the structural and electrical characteristics of 3D twin T-graphene embedded with transition metals (TM). They showed that TM adsorption influences the twin T-electrical graphene’s characteristics. The authors observed the adsorption of Sc, Ti, V, Cr, and Zn on semiconductors, Mn, Cu, and Ni on metals, and Fe and Co on bipolar magnetic semiconductors. Twin T-graphene sheets with TM embedded in them have an energy band gap that narrows with increasing TM atom concentration. The outcomes demonstrated the potential of TM-integrated twin T graphene for usage in electrical and spintronic systems. Twin graphene investigations with the dual doping of Al and Y (Y—B, N, O) atoms at various sites (ortho, meta, and para) were also carried out by Yu L. et al. using first-principle DFT calculations to determine the structural, electrical, and magnetic properties [32]. The AlB-TG system is the most stable dual-doped structure since all Al-Y dual-doped twin graphene (AlY-TG) systems are formed through exothermic methods that result in stable dual-doped structures. In the situations of AlB and AlN doping, dual doping controls the bandgap of twin graphene. The bandgap of twin graphene is controlled by doping when AlB and AlN are utilized, and the pure twin graphene has a direct bandgap, is nonmagnetic, and is semiconductive. By dual doping Al and Y (Y—B, N, O) atoms, the twin graphene’s electrical and magnetic characteristics can be altered. A theoretical foundation for using twin graphene in nanomagnets and spintronic devices is provided in this paper. Graphene nanoribbons (GNRs) can become the fundamental components of spintronic devices thanks to the embedded spin chains. Magnetism in GNRs is typically linked to highly reactive, hard-to-produce localized states along zigzag edges. Friedrich N. et al. showed that by creating atomically precise engineering topological flaws in the interior of the GNR, magnetism can also be induced away from the physical zigzag edges [33]. Two spin-polarized boundary states are created around a pair of substitutional boron atoms that are introduced into the carbon backbone, breaking the conjugation of their topological bands. Electrical transport experiments using boron-substituted GNRs suspended between the tip and the sample of a scanning tunnelling microscope revealed the spin state. A magnetic ground state is made possible by adding two boron atoms to the carbon lattice of graphene nanoribbons (GNRs) as shown in Figure 4.

Osman W. and their colleagues use density functional theory to examine the electrical and magnetic properties of armchair-hexagonal (AHEX) and zigzag-triangular (ZTRI) graphene quantum dots doped with alkali metals [35]. Alkali metals (Li, Na, and K) act as dopants by substituting one of the C-atoms present inside the flake in various locations. Also, the stability of the undoped systems is confirmed using the binding energy. Even though doping makes single-layer structures less stable, bilayer structures have higher binding energy between the layers. Magnetic characteristics are also influenced by stacking; for example, edge pairing causes pure bilayer triangular flakes to turn antiferromagnetic. Doping has a considerable impact on the energy gap; for example, when Na is doped at the upper position in hexagonal flakes, the gap drops from 3.7 eV to 1.5 eV. Zheng Y. et al. showed that designer above-room-temperature magnetic phases and functionalities are abundant in graphene nanomaterials. The capacity to adjust magnetic coupling signs has remained elusive but highly wanted, despite recent confirmation that spins exist in open-shell nanographenes. In atomically accurate open-shell bipartite/non-bipartite nanographene, they have successfully illustrated an engineering method for magnetic ground states by combining scanning probe techniques with mean-field Hubbard model calculations [36]. By disrupting the bipartite lattice symmetry of nanographene, the magnetic coupling direction between two spins was regulated. Additionally, by carefully adjusting the overlap of two spins’ spin densities, the exchange interaction strength between them was broadly controlled, yielding a significant exchange interaction strength of 42 meV.

#### 3.1.2. Logic Application:

Similar to the implantation of biological logic-gates [37,38,39], graphene nanoflakes (GNFs) have recently gained significant interest due to their use in spintronic devices [40,41]. They allow for the induction of magnetism via boundary states, defects, doped magnetic atoms, or strain. In a current study, the authors suggested a number of binary (two-qubit) logic gates in the p-conjugated rhombic graphene nanoflakes (Co_4_-GNF). For the Co_4_-GNF structure, the spin manipulation procedure was completed with a fidelity above 96% in the subpicosecond time scale. They used the position and direction of the spin as the information bits of the binary gates because the spin density of the system continues to be highly localized in some electrical states. Numerous options result in various reorganizations of various classical and quantum logic gates. Both traditional (OR, AND, NAND) and quantum (CNOT, SWAP) binary logic gates are built by carefully mixing the various spin-dynamics processes attained in the Co_4_-GNF structure [27]. A molecular spin logic gate was designed by Zhang W. et al. using two Mn porphyrins connected by a six-carbon monoatomic chain (diMnPh) and sandwiched between two electrodes made of armchair graphene nanoribbons (AGNR). The researchers’ findings demonstrate that the spin-resolved transport features can be successfully controlled by simultaneously modifying the molecules’ initial spin polarizations and zigzag edges. The diMnPh molecular junction can realize various spin logic gates, such as YES, NOT, XOR, OR, and NOR, which are crucial for designing and implementing high-performance and multifunctional molecular spintronic devices in the future. These gates can be realized based on different spin-dependent current-voltage characteristics [42]. The work by the Han Zhou group thoroughly examines the spin couplings between transition metal atoms doped on graphene and demonstrates how they may be used to create various logic gates for spintronic device design [43]. The spin-coupling effect can manifest a certain distance dependence and space propagation, as further confirmed by the impacts of the number of carbon layers and the distance between doped metal atoms on the logic gate implementation. The accomplishments in this work reveal the potential utility of graphene materials and are anticipated to open new research directions for investigating their use in the creation of complex spintronic devices. In this work, the six logic gates AND, OR, NOT, XOR, NOR, and INHIBIT are implemented.

### 3.2. Carbon Nanotubes

Since Iijima’s 1991 discovery of CNTs, there has been a lot of interest in CNT-based spintronics [12]. Different structural categories are used to categorize CNTs. Single-walled carbon nanotubes (SWCNTs) are created by rolling a graphene nanoribbon into a cylinder; multi-walled carbon nanotubes are created by nesting several SWCNTs with various radii together (MWCNTs). Depending on their diameters and chiralities, SWCNTs can have metallic or semiconducting electrical characteristics. MWCNTs can be thought of as a collection of coaxial SWCNTs.

#### 3.2.1. SWCNTs

The design of magnetic materials, spintronic devices, and others are all inextricably connected to the spin-polarized behavior of electrons [3,44]. The research of wang J. et al. on cap-(9, 0) CNTs investigates a defect for adsorbed atoms that could lead to spin polarization on the surface of C30 [45]. To investigate it, they employed the first-principles DFT approach [46,47,48]. The outcomes of the calculations demonstrate that the C adatom drives the formation of spin density and that the distribution of spin density varies for various adsorption positions. The research on spintronic injection and other devices will benefit from this effort. Their research suggests that the asymmetric structure’s spin polarization can be modified by adding adatom defects. The capacity of spin crossover (SCO) molecules to change their spin state in response to various stimuli makes them interesting candidates for nanoscale magnetic switches. Strong Fe-based SCO molecules have proven to be enclosed within the 1D cavities of single-walled carbon nanotubes by Villalva J. et al. [49] as shown in Figure 5. They discovered the SCO process holds up to individual heterostructures being enclosed and placed in nanoscale transistors. A substantial conductance bistability is triggered through the host SWCNT via the SCO switch in the guest molecules. Additionally, unlike crystalline samples, the SCO transition occurs at higher temperatures and exhibits hysteresis cycles, which results in the memory effect. Their findings show how SCO molecules can be processed and positioned into nanodevices using SWCNTs as a support structure, which can also help to adjust their magnetic characteristics. Using DNA strands to helical functionalize carbon nanotubes, the chirality-induced spin selectivity (CISS) effect can polarize carrier spins.

A fundamental comprehension of this effect, which M. W. Rahman and colleagues have studied, is essential for the prospective application of this system in spintronic devices [50]. Due to DNA functionalization, the conduction mechanism was discovered to operate in the strongly localized regime, and the observed magnetoresistance is a result of interference between the forward and backward hopping paths. According to estimates, CISS-induced spin polarization increases the carrier localization length by an order of magnitude in the low temperature range and has a non-trivial impact on the magnetoresistance effect that was not seen in traditional systems.

To create high-performance nanoelectronic devices in the future with minimal interface contact resistance barriers and micro superconducting spintronic devices, C/BN heteronanotubes with polar discontinuity have been created by Wang Y. [51]. Linear interfaces between CNTs and h-BN nanotubes (BNNTs) within heteronanotubes introduce unexpected electrical and spintronic properties that are distinct from those of the individual parts. Polar discontinuities occur at the interfaces of heteronanotubes with zigzag boundaries (ZCBNNTs), creating an inherent electric field that acts on the segment CNTs. To make up for the potential difference, the electric field then induces a charge transfer between the two interfaces; as a result, the interface states (or boundary states) exhibit novel behaviors. For instance, ZCBNNTs are half-metallic in nature. Due to the compensatory effects of the free charge to the polarization charge, the ZCBNNTs’ length and diameter can also be used to alter the electrical properties in an efficient manner. Chen M. et al. developed innovative nanomaterials for use in electrical atomic switches, optoelectronic, and spintronic devices [52]. They have shown that chromium atoms put between parallel SWNT sidewalls can transfer electrons to the benzene rings of the nanotubes by forming hexahapto bonds with the benzene atoms. This maintains the nanotube’s conjugated electrical structure. An appealing method for the reversible chemical engineering of the transport characteristics of aligned carbon nanotube thin films is to link the graphitic surfaces of carbon nanotubes with transition metal atoms, which boosts the transverse conductivity of connected and aligned SWNTs by ≈2100%. They show that a SWNT-aligned device can transition back and forth between a state of high electrical conductivity. They have shown that a device made of aligned SWNTs is capable of being reversibly switched from a state of high electrical conductivity (ON) via light to a state of low electrical conductivity (OFF) by applying voltage. CNTs are excellent choices for supercapacitor electrodes, both when combined with other electrode materials and when used alone. Activating the CNT walls and/or tips will increase the specific surface area. For instance, Pan et al. increased the specific surface area of SWNTs through electrochemical activation from 46.8 m^2^/g to 109.4 m^2^/g, resulting in a three-fold increase in the specific capacitance [53]. For extremely pure SWNTs, Hata and colleagues reported a specific surface area of 1300 m^2^/g [54]. Chen X. et al. have reported energy densities up to 94 Wh/kg (or 47 Wh/L) and power densities up to 210 kW/kg (or 105 kW/L) using an organic electrolyte (1 M Et4NBF4/propylene carbonate) to assure a high voltage of 4 V [55].

#### 3.2.2. MWCNTs

Danilyuk A.L. and colleagues have explored the indirect exchange coupling mediated by conduction electrons in multiwall carbon nanotubes (MWCNTs) with single-domain ferromagnetic nanoparticles (FNPs) implanted inside. They have demonstrated that the static spin susceptibility can spread up to tens of micrometers by modifying the Klinovaja–Loss (KL) model for single-wall CNTs [56]. The adjustment of the Fermi level to the gap created by the spin–orbit interaction is the primary requirement for the long-range exchange interaction (SOI). The suggested method enables measuring the exchange interaction’s energy between FNPs that are part of the same CNT. The results obtained present promising prospects for the fabrication and application of MWCNT-based spintronic devices. Mosse I.S.’s work with nano-tweezers is an important source for the creation of device components that could be useful for quantum information technology [57]. A controlled synthetic chemical technique was used to improve magnetic interactions along nanotube walls. This technique is based on a two-step process that first examines the functionalization of nanotubes (carbonyl groups) and then looks at the attachment of an organo–metallic complex to the carbonyl group. Depending on the functionalization method, mesoscopic electron spin correlations have been seen as well as a distinct transition from superparamagnetism to weakly ferromagnetism. Next, they used a nano-tweezer made from a memory metal alloy to illustrate a novel production method based on nanointegration. The developed devices had quantum rings, crossed junctions, and fine network topologies that can be controlled by nano-probes.

### 3.3. Carbon Nitride

#### 3.3.1. General

A brand-new group of carbon-based materials made up of nitrogen (N) and carbon (C) atoms are known as 2D carbon nitride nanosheets. A recent study has focused heavily on the 2D carbon nitride family due to evidence of its physical, chemical, and morphological features [58,59]. It is interesting to note that by adjusting the N/C ratio and moving the N and C atoms around in the lattice structure, the electric characteristics of carbon nitride nanosheets can be flexibly designed to range from half-metal to semiconductor [60,61], offering insightful design guidance for novel g-C3N4-based two-dimensional elastic electrical and spintronic devices. Biaxially strained graphitic carbon nitride’s (g-C3N4) mechanical and electrical characteristics were studied by Qu L.-H. and colleagues [62]. The results demonstrated that g-C3N4 has substantial linear elasticity and extremely isotropic mechanical characteristics. The photon transition between band gaps was modest, implying that the g-C3N4 monolayer is not a good material for solar cells. It was discovered that the spin-unrestricted band gap of g-C3N4 can be over-estimated, and that sufficient biaxial strain can cause the spin splitting of g-C3N4. The experimental implementation of bare C3N nanoribbons spintronic devices is challenging. This is because edge reconstruction will take place in the bare C3N nanoribbons. Utilizing ferromagnetic Ni electrodes, Zeng J. and Zhou Y. thoroughly examined the spin-polarized characteristics of 2D C3N sheets and the related hydrogen-passivated nanoribbons [63]. On top of the Ni electrodes, a 2D C3N sheet exhibited semiconductor-to-metallic spin filtering and a positive magnetoresistance effect. Particularly impressive were the increased spin polarization effectiveness and the appearance of the negative magnetoresistance effect following the transformation of the 2D C3N sheet into hydrogen-passivated nanoribbons. These findings suggest that C3N has a lot of promise for use in nanoscale spintronics.

#### 3.3.2. Doping

In addition to using carbon nitride in its purest form, doping the monolayer with atomic impurities has also been investigated to enhance its physical and chemical properties. By manipulating the band structure, doping with non-metal atoms such as S, O, B, and C can increase the lifetime of photogenerated electrons and the range of absorbed light wavelengths [64]. One efficient and convenient approach to improvement is co-doping carbon nitride nanosheets. According to research by Bafekry et al., co-doping changes the semiconducting properties of C6N7 into a Dirac half-metal with a 2.3 eV band gap in the spin-up channel, and a gapless Dirac conic band structure in the spin-down channel. These results show that the 2D carbon nitride family has applicability in spintronic devices, made possible by the co-doping technique, in the band structure [61]. The metal-adsorbed C4N systems are remarkable potential candidates for the development of electrical and spintronic devices. Liu W. and colleagues presented a novel possibility for controlling the ferromagnetism in light element systems [65], by adding ferromagnetism to materials with only s and p electrons in light elements. In this study, the ferromagnetic properties of carbon-doped boron nitride (B-C-N) nanosheets were investigated. These nanosheets were produced by the high-temperature annealing of a stacked mixture of boron nitride nanosheets (BNNSs) and graphene that had been prepared by urea-assisted aqueous exfoliation. The BCN nanosheets showed strong ferromagnetic responses, with a saturation magnetization of 0.142 emu/g at ambient temperature. They discovered that the ferromagnetic characteristics of the BCN nanosheets are significantly influenced by the thickness of the predecessors.

The electrical and magnetic characteristics of the 2D materials were successfully altered by surface adsorption. Xu M. et al. used the density functional theory to examine the adsorption behaviors of 16 metal atoms that were adsorbed on the 2D dumbbell C4N (DB C4N), including alkali metals, alkaline earth metals, 3D transition metals (TMs), and precious metals [66]. The alkali metals (Li, Na), alkaline-earth metals (Be, Mg), 3dTMs (Ti, V, Cr, Mn, Fe, Co), and precious metals (Ag, Pt, Au) were shown to open the zero bandgaps of DB C4N through the charge transfer between the adatoms and the surface of the material. Additionally, the monolayer C4N was made magnetic using the 3D transition and precious metal adatoms.

### 3.4. Fullerenes

We must be able to comprehend and manipulate metal-organic interactions for the functionalization of organic complexes for next-generation electrical and spintronic devices. The so-called single molecular magnets (SMM) are of special relevance for magnetic data storage applications because they provide the opportunity to store information on a molecular scale. The research of Seidel J. et al. focused on the adsorption characteristics of the archetypal SMM Sc3N@C80 produced as a monolayer film on the Ag(111) substrate [67]. They offered convincing proof that the adsorption on the Ag(111) surface caused a pyramidal deformation of the otherwise planar Sc3N core inside the carbon cage. It is possible to link this adsorption-induced structural change in the Sc3N@C80 molecule to a charge transfer from the substrate into the lowest unoccupied molecular orbital of Sc3N@C80, which drastically changes the charge density of the fullerene core. This research demonstrated how such an indirect interaction between the metal centers of SMMs that are enclosed and the metal surfaces can significantly modify the geometric shape of the metallic centers, potentially changing the magnetic characteristics of SMMs on surfaces as well.

### 3.5. Others

Carbon-based materials have received considerable and widespread research attention ever since monolayer graphene was successfully synthesized. They show great promise for use in electronic devices, even replacing silicon-based electronics, optoelectronics, and spintronics, due to their outstanding transport capacity and conductivity. In work by Liu C. et al., amorphous carbon nanosheets (CNs) with an average of 3.6 nm graphene nanocrystals inside were found to macroscopically attain room-temperature ferromagnetic ordering [11]. The magnetization that could result from the ferromagnetic coupling between the zigzag edges of nanocrystals in CNs would be substantial (>0.22 emu/g), which is two orders of magnitude more than what has been observed in defective graphite. Additional experiments and first-principles calculations showed that the zigzag edges’ separation might effectively influence the magnetic coupling in CNs.

In another study, M.A.M. Keshtan and M. Esmaeilzadeh investigated the topological and spin-dependent electron transport characteristics of a trans-polyacetylene molecule [68]. Even though their Hamiltonians do not adhere to chiral symmetry, it was discovered that molecules with intracellular single carbon–carbon bonds and an even number of monomers in their chains exhibit two edge states and contain topological features. The quantum spin-dependent electron transport features are induced and manipulated using two perpendicular and transverse electric fields and a perpendicular exchange magnetic field. In distinct electron energy zones that are extended by stronger exchange fields, the exchange field causes spin polarization. As a result, the suggested gadget functions perfectly as a spin filter. A perfect spin caloritronics device with a carbon-based organic chain was proposed by Tan F. et al. It is possible to achieve a spin-semiconducting feature that results from edge localized states [69]. A significant spin Seebeck coefficient emerges from the spin-dependent transport gaps. Moreover, at room temperature, the dimensionless spin thermoelectric figure of merit (FOM) can be increased to 35. As a result of a temperature difference, it is also possible to produce a pure spin current or single-spin current at some chemical potentials, and the chemical potential can also be used to control the transport directions of these currents. Optically addressable spins, which combine a long-lived qubit with a spin-optical interface for external qubit control and readout, are a promising framework for quantum information science. A modular qubit architecture is made possible by the ability to chemically synthesize such systems, which can produce optically addressable molecular spins. This architecture can be transported between different environments and atomistically customized for specific applications through a bottom-up design and synthesis. D. D. Awschalom and colleagues show how manipulating the host environment can affect the spin coherence in such optically addressable molecular qubits [70]. They create noise-insensitive clock transitions utilizing chromium (IV)-based molecular qubits in a nonisostructural host matrix through a transverse zero-field splitting that is not possible with an isostructural host. In a nuclear and electron spin-rich environment, this host-matrix engineering produces spin-coherence times of more than 1.0 µs for optically addressable molecular spin qubits. Their findings show how a customizable molecular platform may be used to evaluate qubit structure–function relationships and highlight potential applications for employing molecular qubits for nanoscale quantum sensing in noisy environments.

### 3.6. Doped and Codoped

The properties of carbon materials can be enhanced or modified through the use of adatoms, adsorption, doping, defects, the addition of an electric field, changes in tension, and other techniques. Based on the density functional theory, Zhongyao Li and Min Chen investigated the potential half-metallic behavior in three-dimensional, transition metal (Fe, Co, and Ni)-decorated two-dimensional polyaniline (C3N). The Ni-decorated polyaniline ((C3N)2Ni) is a nonmagnetic semiconductor with an increased band gap, according to the estimated electronic structures, but the Fe and Co decorated polyanilines ((C3N)2Fe and (C3N)2Co) are magnetic half-metals. The spacing between 3d transition metal atoms and C3N can change the energy windows and band gaps. 3d-transition-metal-adorned C3N can be used in nanoscale spintronic devices because of its wide half-metallic energy window and suitable band gap. Xia B. et al. established a method for introducing magnetism into carbon nanosheets using a single Cr cation that is only attached to two-dimensional carbon nanosheets via Cr-N bonds [71]. The highest magnetization (Cr: 2.0%, 0.86 emu g^−1^) under 3 T was achieved at 50 K, where the magnetization changes with the Cr concentration. In the samples, it was shown that the anchoring of Cr can cause paramagnetism and ferromagnetism, and that the magnetization is strongly correlated with the Cr level. A method to create a magnetic carbon matrix was presented in this study, which lays the groundwork for several potential spintronics device applications to carbon-based, low-dimensional materials. A third-generation semiconductor material with a wide band, silicon carbide (SiC), is important in the power and electronics industries. SiC has more than 200 distinct polytypes or crystal lattice alteration patterns. One of the significant polytypes [72] that has received a lot of attention is 4H-SiC. Long Lin’s group has thoroughly studied the electronic structure, magnetic, and optical properties of 4H-SiC doped with a single V atom, single Fe atom, and (V, Fe) co-doped 4H-SiC [73]. The 3d orbitals of V or Fe can contribute significantly to the single V or Fe dopant’s ability to introduce magnetism in pure 4H-SiC. The coupling relationship between the V-3d, C-2p, and Fe-3d states is believed to be the cause of ferromagnetism, and the (V, Fe) co-doped system prefers FM states. Red-shift phenomena emerge in the absorption spectrum as a result of the addition of V and Fe atoms, greatly enhancing the absorption strength in visible light. The outcomes demonstrate that the (V, Fe) co-doped 4H-SiC system will offer a potential method for the advancement of spintronic devices and optical applications in the future. Table 1 summarize some important carbon- based material spintronic applications.

## 4. Spin Relaxation in Carbon-Based Materials

The conduction electron spin lattice relaxation time (CESR), T_1_, is the typical time for a spin system that has been driven out of equilibrium by, for example, a microwave field at electron-spin resonance ESR or a spin-polarized current to return to thermal equilibrium. A sufficiently long spin lifetime is necessary to be used in “spintronics” devices, in which electron spins are used to process information [74]. In the MgB2 superconductor, Simon F. and colleagues measured the spin lattice relaxation time T_1_ of conduction electrons as a function of temperature and magnetic field [75]. During electron-spin resonance conditions with amplitude-modulated microwave stimulation, researchers employ a technique based on the detection of the z component of the conduction electron magnetization. Despite the considerable CESR line broadening caused by irreversible diamagnetism in the polycrystalline sample, a lengthening of T_1_ below Tc (critical temperature) is seen. They can measure the individual contributions to T_1_ from the two different forms of the Fermi surface because of the field independence of T_1_ for 0.32 and 1.27 T. A phase-pure crystal of potassium-doped p-terphenyl, [K(222)]2[p-terphenyl3], is isolated by Gadjieva N.A. et al. [76]. Magnetometry and electron spin resonance (ESR) are used to conduct in-depth research on the emerging antiferromagnetism in the anisotropic structure. The antiferromagnetic coupling in this system, which occurs in all three crystallographic directions, has been described by the authors using these experimental findings in combination with calculations using the density functional theory. The terphenyls’ ends, where an extra electron on nearby p-terphenyls antiferromagnetically couples, showed the strongest coupling. These results suggest that potassium-doped p-terphenyl exhibits magnetic fluctuation-induced superconductivity, which is closely related to high Tc cuprate superconductors.

Using liquid ammonia, Markus B.G. et al. reported synthesizing few-layer graphene (FLG) doped with Li and Na. Chemical exfoliation was used to prepare the FLG material [77]. The appearance of powerful, metallic-like electron spin resonance (ESR) modes and the modification of the Raman G line into a Fano line shape are evidence of the high concentration of graphene doping in liquid ammonia for both types of alkali atoms. The spin-relaxation duration in the materials was 6–8 ns, which is equivalent to the longest values discovered in spin transport tests on ultrahigh-mobility graphene flakes. This time was calculated from the ESR line width. This might make this substance a good contender for spintronics devices. However, since sodium is a very common metal, a successful sodium doping attempt might be a promising replacement for lithium batteries.

The study by Ren-Shu Wang et al. provides a complete set of parameters through a thorough investigation on a K_3_C_60_ sample that has been well-characterized [78]. K_3_C_60_ is a promising three-dimensional superconducting magnet material with the advantage of the rich carbon abundance on the Earth due to the high upper critical field of 33.0 ± 0.5 T obtained from the direct electrical transport measurements, along with the relatively high critical temperature and large critical current density. The examination of all independently acquired parameters points to the peculiar nature of K_3_C_60_’s superconductivity, including contributions from electron correlations and electron–phonon coupling. By using the ESR technique, paramagnetic centers in heterofullerides with the compositions A_2_MC_60_ and AM_2_C_60_ (A = K, Rb, M = Mg, Be) were studied by Kytin V.G. and his collegues [79]. The ESR signal was discovered to be composed of two lines with distinct temperature dependences in terms of ESR absorption magnitude. This provides proof that there are at least two different types of paramagnetic centers present. The first type’s centers exhibit localized spin behavior, whereas conduction electrons can be used to describe the remaining centers.

The electron spin resonance (ESR) signal of undoped and potassium-doped SWCNTs was investigated by Galambos M. et al. [80]. They identify the signals of the conduction electron spin resonance (CESR), the low intensity impurity, and the superparamagnetic background. Only the alkali atom doping causes the latter to be present. They critically evaluate the possibility that the CESR signal could be residual graphitic carbon, which they categorically rule out, in order to identify it. They provide precise values for the signal intensities, related spin concentration, and g factors. The density of states on the SWCNT assembly can be calculated using the CESR signal intensity.

## 5. Organic Spintronics

Carbon-based, molecular, or polymeric semiconductors are used in organic electronics because they are inexpensive to produce, mechanically adaptable, and, most critically, have practically limitless and chemically tunable electrical and optical properties. The organic light-emitting device (OLED), which is utilized in displays, is the most widely used commercial organic electronic application (notably in smart phones). By combining organic materials into spintronics and spin dependent effects in organic electronics, organic spintronics strives to merge these two fields. According to Dediu et al. devices with ferromagnetic La0.7Sr0.3MnO3 (LSMO) electrodes and a sexithienyl (6T) spacer exhibit room-temperature magnetoresistance [81]. In order to create lateral devices, LSMO films were shaped into electrodes separated by a small gap (100–500 nm). The gadget’s resistance had a distinct magnetic field dependence. The initial investigations on organic spin valves were motivated by this work and used ferromagnetic LSMO and Co electrodes in a vertical layer stack (layers deposited on top of each other) separated via an Alq3 spacer (thickness 100–200 nm). Utilizing the unique material characteristics of organic semiconductors for spintronic applications is gaining popularity. In a small molecule system based on dinaphtho[2,3-b:2,3-f]thieno[3,2-b]thiophene (DNTT), Angela Wittmann et al. investigate the application of a pure spin current from Permalloy at ferromagnetic resonance [82]. They are able to systematically examine the influence of interfacial characteristics on the spin injection efficiency via molecular design thanks to the unique tunability of organic materials. We demonstrate that the interfacial molecular structure and side chain substitution of the molecule allow delicate tuning of the spin injection efficiency at the interface as well as the spin diffusion length. Organic photovoltaics (OPVs) based on non-fullerene acceptors have received a lot of interest in the past 10 years because of their excellent potential to achieve high-power conversion efficiencies. The primary difficulties in permitting effective charge separation/transport and a low voltage loss simultaneously are what limit the development of higher performance OPVs. To match the commercially available polymer PM6, Yuan J. et al. have designed and created a new type of non-fullerene acceptor, Y6, that uses an electron-deficient, core-based central fused ring with a benzothiadiazole core [83]. With both conventional and inverted architecture, the Y6-based solar cell achieves a high-power conversion efficiency of 15.7% using this method. By doing this study, we offer fresh perspectives on how to build new non-fullerene acceptors to achieve increased photovoltaic performance in OPVs by utilizing the electron-deficient, core-based central fused ring.

## 6. Challenges in Carbon-Based Spintronics

Twisteronics is currently a popular issue in the study of 2D materials. Although the superconductivity in twisted graphene systems appears to be of an uncommon type, the underlying theoretical description is still elusive, which poses a significant issue for carbon-based spintronics. Increasing production rate, scalability, homogeneity, and quality control are problems for the industrial-scale synthesis of carbon-based spintronic materials. In general, varying quality across producers is a problem, and it is imperative to improve a single standard or grading system. Improving the quality and reproducibility of patterned nanostructures, effectively coupling light into and out of graphene, and expanding plasmon tunability to the vis-NIR range are challenges in plasmonic and optical qualities. Developing regulated fabrication processes that result in reliable and repeatable devices is another problem in electrical and spintronic applications. Of course, there are also significant challenges with scalability and wafer-scale integration. Nanotube-based electronics have two main difficulties that must be overcome. Connectability is one of the difficulties; while it is one thing to construct a single nanotube transistor, it is quite another to connect millions of them. A single component at a time is often used in the current techniques of nanotube electronics, which is not practical. The suggested theoretical research and experimental findings typically differ significantly, since the experiments are so complex. These disagreements suggest that elaborate physical mechanisms and precise theoretical models must be established to effectively direct research efforts and interpret experimental findings. The limitation in material science with carbon-based spintronics is that the majority of materials with thicknesses close to the atomic level are temperature-, oxygenation-, and moisture-sensitive. They need to operate above room temperature and be stable in the presence of air.

## 7. Outlook and Future Perspective

Due to their unique spin-dependent characteristics, such as lengthy spin relaxation durations, long diffusion lengths, and high spin–orbit coupling, carbon-based materials have received a lot of attention in the field of spintronics. Because of its high charge carrier mobility, long spin lifetime, and long diffusion length, graphene has distinguished itself as a superior platform for future spintronic devices. Effective spin logic and non-volatile data storage are made possible by the tunable bandgap and robust spin–orbit coupling that carbonitrides show. The integration of spintronics and photonics into a single platform for both light-based and spin-based quantum computing is the goal of ongoing research and development. Researchers hope to be able to control the electron spin dynamics in nanostructured 2D magnetic materials activated using brief laser pulses by employing specially created photonic circuits. The rapidly developing field of spintronics, which has contributions from a wide range of nations and disciplines, including biology, chemistry, physics, electrical engineering, computer science, and mathematical information theory, promises to make fundamental discoveries in both pure and applied science as well as have a significant impact on future technology. In contrast to traditional charge-based information processing technologies, using spin as a state variable in logic devices has various benefits, including non-volatility, faster and more energy-efficient data processing, and higher integration densities. Spintronic computing has the potential to meet the constantly rising performance requirements of upcoming abundant-data applications.

## Figures and Tables

**Figure 1 nanomaterials-13-00598-f001:**
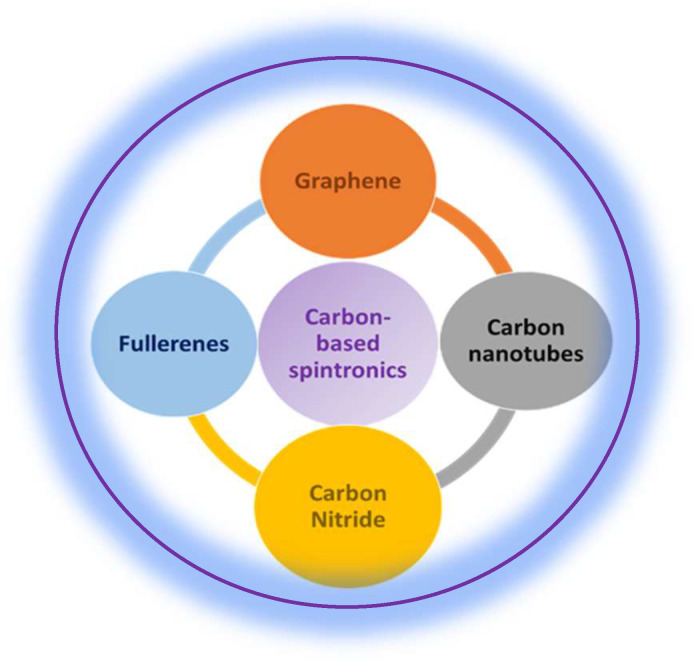
Schematic representation of carbon-based spintronics.

**Figure 2 nanomaterials-13-00598-f002:**
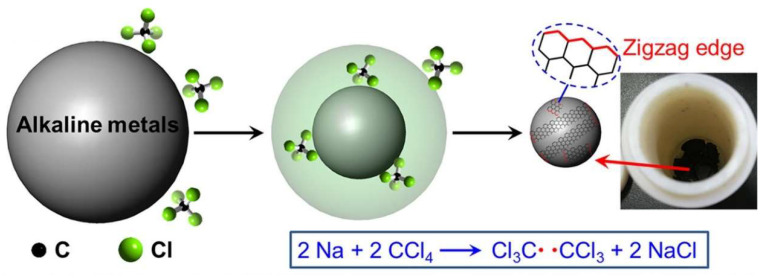
Synthesis of ferromagnetic carbon nanosheets. Reproduced from ref [11] with permission of Journal of Physical Chemistry C copyright 2020.

**Figure 3 nanomaterials-13-00598-f003:**
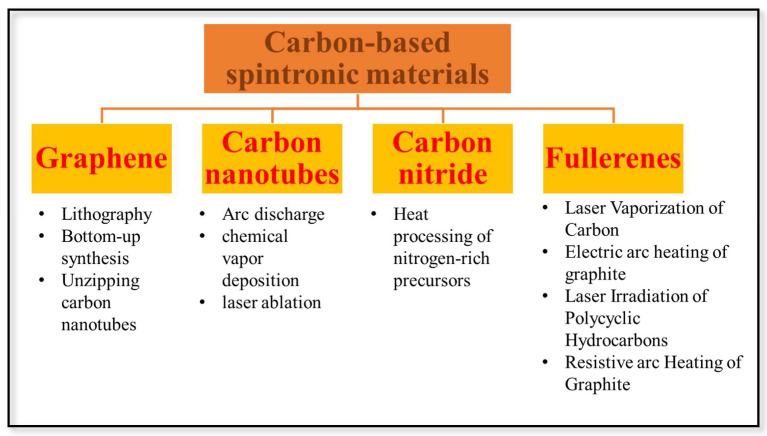
Schematic representation of the preparation of carbon-based spintronic materials.

**Figure 4 nanomaterials-13-00598-f004:**
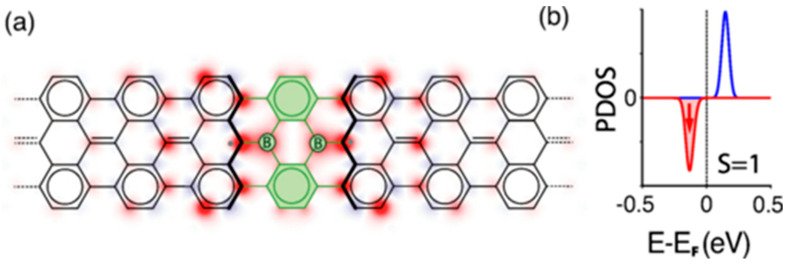
(**a**) Structure of the 2B-7AGNR shown over a color map representing the spin polarization density map, computed using density functional theory simulations ([16]) (green represents the boron moiety). (**b**) Spin-resolved projected density of states (PDOS) over carbon atoms around the boron dimer. Net spin polarization of one kind confirms the ferromagnetic alignment of the two magnetic moments. Reproduced from ref [33] with permission of Phys. Rev. Lett. copyright 2020. The production of nitrogen-doped porous graphene nanoribbons (N-GNRs) on Ag(111) was accomplished by Pawlak R. in a different study by using a silver-assisted Ullmann polymerization of brominated tetrabenzophenazine [34]. Combining scanning tunneling microscopy (STM), atomic force microscopy (AFM) with CO-tip, scanning tunneling spectroscopy (STS), and density functional theory provides insights into the hierarchical reaction pathways from single molecules toward the formation of one-dimensional organometallic complexes and N-GNRs (DFT).

**Figure 5 nanomaterials-13-00598-f005:**
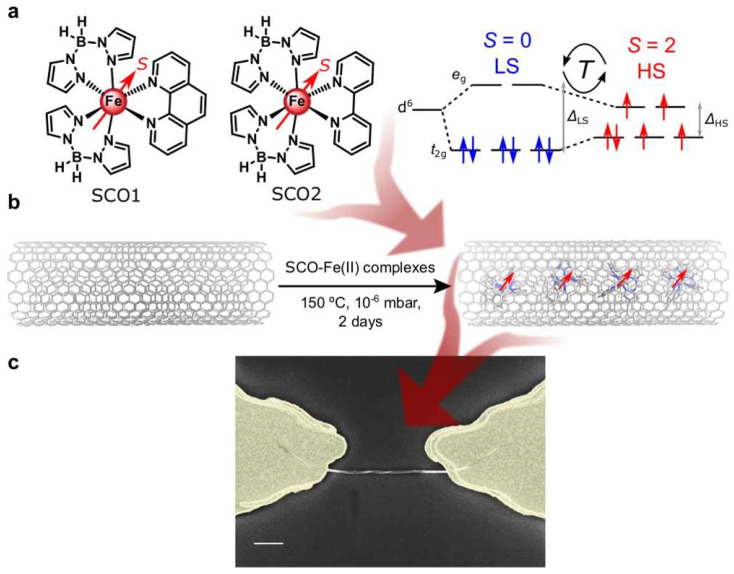
(**a**) Complex SCO1, [Fe(H_2_Bpz_2_)_2_(phen)] and SCO2, [Fe(H_2_Bpz_2_)_2_(bipy)] encapsulated in this work. Spin (S) level distribution in the high (HS) and low (LS) spin states of the molecules. (**b**) Schematic pathway for the encapsulation of Fe (II) SCO complexes in oSWCNTs, resulting in SCO@SWCNT. (**c**) Scanning electron microscopy (SEM) image of a transistor-like device containing a SCO2@SWCNT heterostructure trapped via dielectrophoresis. Scale bar: 500 nm. Reproduced from ref [49] with permission of Nat. Commun. copyright 2021.

**Table 1 nanomaterials-13-00598-t001:** Summary of carbon-based spintronic application.

Carbon-Based Material	Application	References
Carbon nanosolenoid (CNS)	Riemann surfaces	[15]
Partially hydrogenated graphene (PHGr)	Single atom catalysts	[16]
Five-membered ring-decorated nanographene	Overcurrent protection	[29]
Zigzag-edged graphene nanoribbons (ZGNRs)	Spin-filtering device	[32]
Zigzag-edge graphene nanoribbon (ZGNR) electrodes	Spin filtering and the negative differential resistance (NDR) effect	[35]
Boron atoms (B2) doped 7-atom-wide armchair graphene nanoribbons (B2-7AGNRs)	Pie magnetism and spin-dependent transport	[28]
Graphene lattice with transition metal atom	High magnetic anisotropy energy (MAE)	[17]
Manganese porphyrin molecules connected to graphene electrodes	Multifunctional spin logic gates	[42]
Metal-doped graphene	Logic gate application	[43]
Metal embedded twin T graphene	Bipolar magnetic semiconductor for Fe and Co	[30]
DNA-functionalized carbon nanotubes	Spin filters	[50]
Carbon nanotubes (CNTs)	Spintronic injection	[45]
Carbon nanotubes (CNTs) with embedded ferromagnetic materials	Indirect exchange coupling	[52]
Spin cross over SWCNT	Nanoscale magnetic switches	[49]
Carbon nitride C6 N7 atomic doping	Two-dimensional Dirac half-metal	[61]
C3N	Semiconductor-to-metallic spin filtering	[63]
Carbon-doped boron nitride nanosheets	Manipulation of ferromagnetism in light element systems.	[65]

## Data Availability

All data are available upon email request. Restrictions apply to the availability of these data. Some data are not publicly available since some articles are not open access.

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
