# Peer review of "Recent Advances in the Spintronic Application of Carbon-Based Nanomaterials"

_nanomaterials, 2023, doi:10.3390/nano13030598_

Round 1
Reviewer 1 Report
This review is moderately interesting. First, a review with only 60 some references falls a bit short of expectation. The redeeming value of this review is the frequent reference to articles published very recently, some in 2022. That is good, but of short value in time.
As the authors speak in terms of "discovery" when reporting DFT calculations, I had to check every reference to find out if it was theory, simulation or experiment. The review would be stronger if the three aspects were not blended together. Or else, since the review is mostly about theory, make the point to specify when you report on experiments. The review does not contain any theoretical presentation, so this review is roughly like the "digest" that I would consult in my early years in research. Overall, the review is only moderately interesting, but it could be improved by sharpening the text, possibly adding references, as I will suggest below when I go through the text line by line.
lin 38 : I propose to stop playing the old tune of "organics are cheaper". Making millions of interconnects to nanoribbons produced by e-beam lithogarphy is not going to be cheap.
41: Could the authors point to a paper where spintronics develops capacitors. Are they thinking of the work of Barnes et al?
114 : "discover" refers to DFT calculations. I would use a softer term
121 : about magnetic edge states, I know of other references,
Angewandte Chemie International Edition, 59(29):12041{12047, 2020
Chem Asian J, 15(22):3807{3811, 2020
Journal of the American Chemical Society, 142(29):12568{12573, 07 2020
so I suspect there are many more and a review article should attempt to cover more works that their restricted selection, or else, they have to explain why they dismiss all others.
129: I doubt that Einstein theory of relativity is relevant, even if both the curvature of space and the curvature of a graphene sheet both are about "curvature".
150 : intriguing statement about the spin-filtering of the double atomic chain. Could the authors expand on this, with a graph, a filtering value maybe?
170 : explain B2
226+227 : looks like some authors added this part without reading the previous author. There is a repeat, but new references.
233 : why would a CNOT gate be "quantum"
This very short passage on logic does no seem justified, or should be expanded, showing that carbon nanostructures have allowed making spin qubits and then present a whole array of results or attempts to make qubits this way.
271 : one ref at least missing. Why would these authors not mention the tremendous effort of their neighbors who have done a lot to discover and characterize CISS. Including work with DNA strands, single and double, and with polypeptides also.
292 : the sentence with ZCBNNT is cut.
350 : what is BeCeN
375 : under others, why not mention the fantastic results of D. Awschalom ?
392 : reference missing. or is it 62 ?
399 : capitalize "The"
423 : what does Rieman surface mean as an applicationfor nanosolenoid
426: define twisteronics, give examples
435 : agreed, the connectability is yet to be demonstrated.
444: say slow spin lattice relaxation (for example)
444: really ? I read that spin diffusion length is short even though the spin lifetime is long, because the collision length is short and the spin diffusion length is the square root of the product of spin flip mean free path and the collision mean free path. Fro memory, Coey co-authored a paper on this matter. And I heard Fert mention the problem that the spin diffusion length is short and he must have published this.
447 : "effective" meaning ? This statement is interesting but seems to come as an after thought. Can the authors elaborate on the role of a tunable band gap on implementing spin logic, this sounds interesting.
497: incomplete reference
512: remove "cite this"
Author Response
We are thankful to the reviewers for sharing their valuable opinion on our work. The comments truly helped us to improve the content. We tried our level best to answer the comments of the reviewers through this revised form of the manuscript.
Reviewer: 1
Comments
This review is moderately interesting. First, a review with only 60 some references falls a bit short of expectation. The redeeming value of this review is the frequent reference to articles published very recently, some in 2022. That is good, but of short value in time.
As the authors speak in terms of "discovery" when reporting DFT calculations, I had to check every reference to find out if it was theory, simulation or experiment. The review would be stronger if the three aspects were not blended together. Or else, since the review is mostly about theory, make the point to specify when you report on experiments. The review does not contain any theoretical presentation, so this review is roughly like the "digest" that I would consult in my early years in research. Overall, the review is only moderately interesting, but it could be improved by sharpening the text, possibly adding references, as I will suggest below when I go through the text line by line.
Response: We thank the reviewer for the valuable and insightful comment and we have addressed all the queries point by point as following. We have been able to come across new references after a thorough search. Now, they are covered in Section wherever needed of the updated paper.
line 38 : I propose to stop playing the old tune of "organics are cheaper". Making millions of interconnects to nanoribbons produced by e-beam lithogarphy is not going to be cheap.
We agree with the comment and removed the line.
41: Could the authors point to a paper where spintronics develops capacitors. Are they thinking of the work of Barnes et al?
We have added spintronics develops capacitors in the section 3.1 graphene based and 3.2a SWCNTs with new references such as:
Zhao, Y.; Hu, C.; Hu, Y.; Cheng, H.; Shi, G.; Qu, L. A versatile, ultralight, nitrogen-doped graphene framework. Angew. Chemie - Int. Ed. 2012, 51, 11371–11375.
Pan, H.; Poh, C.K.; Feng, Y.P.; Lin, J. Supercapacitor electrodes from tubes-in-tube carbon nanostructures. Chem. Mater. 2007, 19, 6120–6125.
Izadi-Najafabadi, A.; Yasuda, S.; Kobashi, K.; Yamada, T.; Futaba, D.N.; Hatori, H.; Yumura, M.; Iijima, S.; Hata, K. Extracting the Full Potential of Single-Walled Carbon Nanotubes as Durable Supercapacitor Electrodes Operable at 4 V with High Power and Energy Density. Adv. Mater. 2010, 22, E235–E241.
Chen, X.; Paul, R.; Dai, L. Carbon-based supercapacitors for efficient energy storage. Natl. Sci. Rev. 2017, 4, 453–489.
We are not thinking of the work of Barnes et al.
114 : "discover" refers to DFT calculations. I would use a softer term
We have changed the term to ‘revealed’.
121 : about magnetic edge states, I know of other references,
Angewandte Chemie International Edition, 59(29):12041{12047, 2020
Chem Asian J, 15(22):3807{3811, 2020
Journal of the American Chemical Society, 142(29):12568{12573, 07 2020
so I suspect there are many more and a review article should attempt to cover more works that their restricted selection, or else, they have to explain why they dismiss all others.
We have added the reference
In the section 3.1 graphene based:
Mishra, S.; Beyer, D.; Eimre, K.; Ortiz, R.; Fernández-Rossier, J.; Berger, R.; Gröning, O.; Pignedoli, C.A.;
Fasel, R.; Feng, X.; et al. Collective All-Carbon Magnetism in Triangulene Dimers. Angew. Chemie - Int. Ed.
2020, 59, 12041–12047.
Keerthi, A.; Sánchez-Sánchez, C.; Deniz, O.; Ruffieux, P.; Schollmeyer, D.; Feng, X.; Narita, A.; Fasel, R.;
Müllen, K. On-surface Synthesis of a Chiral Graphene Nanoribbon with Mixed Edge Structure. Chem. –
An Asian J. 2020, 15, 3807–3811.
In section 3.1a Doping or Embedding
Pawlak, R.; Liu, X.; Ninova, S.; D’Astolfo, P.; Drechsel, C.; Sangtarash, S.; Häner, R.; Decurtins, S.;
Sadeghi, H.; Lambert, C.J.; et al. Bottom-up Synthesis of Nitrogen-Doped Porous Graphene Nanoribbons.
- Am. Chem. Soc. 2020, 142, 12568–12573.
Its beyond our scope to cover all the topics in this review hence we have dismissed some of them.
129: I doubt that Einstein theory of relativity is relevant, even if both the curvature of space and the curvature of a graphene sheet both are about "curvature".
Yes, we agree with the reviewer and have removed the sentence to avoid confusion.
150 : intriguing statement about the spin-filtering of the double atomic chain. Could the authors expand on this, with a graph, a filtering value maybe?
The spin filtering efficiency (SFE) of the P spin configuration is very close to 100%. Additionally, it is evident that Iup starts to decline at a rate of 1.0 (1.0) V after exceeding 0.5 (0.5) V. Iup is therefore muted at high bias, resulting in the NDR effect. Contrary to P spin configuration, AP spin configuration's current is embodied in unidirectional spin-dependent features; therefore, Iup (Idn) only manifests when the negative (positive) bias voltage is greater than 0.3 (0.3) V. This indicates that only spin-down channels are open at positive bias, whereas only spin-up channels are open at negative bias. Therefore, bipolar spin filtering and spin rectification can be seen in the all-carbon device in an AP spin configuration. When the bias value approaches 0.3 (0.3) V, the SFE reaches around 100% (100%).
170 : explain B2
B2 is the pair of substitutional boron atoms (B2)
226+227 : looks like some authors added this part without reading the previous author. There is a repeat, but new references.
The reference 27 describe both the effect of doping and the logic gate application hence its used twice in section 3.1a and 3.1b.
233 : why would a CNOT gate be "quantum"
This very short passage on logic does no seem justified, or should be expanded, showing that carbon nanostructures have allowed making spin qubits and then present a whole array of results or attempts to make qubits this way.
The first qubit in a CNOT gate is typically referred to as the control qubit, and the second qubit is typically referred to as the target qubit. The CNOT gate is expressed in basis states. In the areas of quantum computation and quantum information, CNOT gates are extremely significant. In fact, it appears that CNOT gates and single qubit unitary gates make up a universal set for quantum computation, allowing any possible unitary operation on an n-qubit system to be done using only these two types of gates, single qubit unitary gates and CNOT gates
271 : one ref at least missing. Why would these authors not mention the tremendous effort of their neighbors who have done a lot to discover and characterize CISS. Including work with DNA strands, single and double, and with polypeptides also.
The reference is not missing.
292 : the sentence with ZCBNNT is cut.
We have completed the sentence.
350 : what is BeCeN
Its carbon-doped boron nitride (B-C-N) amd its changed in updated manuscript.
375 : under others, why not mention the fantastic results of D. Awschalom ?
We have added “Bayliss, S.L.; Deb, P.; Laorenza, D.W.; Onizhuk, M.; Galli, G.; Freedman, D.E.;
Awschalom, D.D. Enhancing Spin Coherence in Optically Addressable Molecular Qubits through Host-
Matrix Control. Phys. Rev. X 2022, 12, 031028.” In the section 3.5.
392 : reference missing. or is it 62 ?
It is same reference now changed to 69.
399 : capitalize "The"
We have capitalize “The”
423 : what does Rieman surface mean as an applicationfor nanosolenoid
The authors describe an easy bottom-up synthesis of a 3D -extended nanographene carbon material CNS with high yields that resembles a Riemann surface. In mathematics, Riemann surfaces are distorted copies of the complex plane. Locally, they resemble patches of the complex plane, but on a larger scale, the topology may not be plane-like. It is anticipated that nanostructured graphitic carbon materials with helicoid topology will have intriguing electrical and optical properties.
426: define twisteronics, give examples
Due to an unusual flattening of its lowest energy bands, twisted bilayer graphene exhibits insulating and superconducting phases. The research by Dmitry V. Chichinadze et. al shows because of extraordinary flatness of its lowest energy bands, twisted bilayer graphene exhibits insulating and superconducting phases1. Near half filling of the valence band (n ≈ -2), superconductivity with greatest T c is seen. The measurements demonstrate that the triple lattice rotation symmetry is broken in the superconducting phase in a significant portion of the superconducting dome around n = -2; thus, a superconductor is also a nematic.
435 : agreed, the connectability is yet to be demonstrated.
We have mentioned it as one of the challenges which needed to overcome in future
444: say slow spin lattice relaxation (for example)
Reduced graphene oxide obtained commercially was found to have two components, as shown by the two spin-spin relaxation times T2CW -1. The values T2A -1 = 77 MHz and T2B -1 = 250 MHz were discovered at 290 K; at 80 K, they were T2A -1 = 50 MHz and T2B -1 = 250 MHz, respectively2. The following relaxation periods were noted at 300 K for rGO created via reduction using hydrazine hydrate: T1CW-1 = 180 MHz and T2CW-1 = 186 GHz, respectively3.
444: really ? I read that spin diffusion length is short even though the spin lifetime is long, because the collision length is short and the spin diffusion length is the square root of the product of spin flip mean free path and the collision mean free path. Fro memory, Coey co-authored a paper on this matter. And I heard Fert mention the problem that the spin diffusion length is short and he must have published this.
- Yan et al. discovered that the 70–130 µm Long spin diffusion length in Few-Layer Graphene Flakes is more than previous estimates made with sharp-switching electrodes4.
447 : "effective" meaning ? This statement is interesting but seems to come as an after thought. Can the authors elaborate on the role of a tunable band gap on implementing spin logic, this sounds interesting.
Modern digital integrated circuits (ICs) have been using silicon-based complementary metal-oxide semiconductors (CMOS) as their primary logic architecture for decades, but this technology will soon reach its performance threshold. As a result, numerous studies have been conducted utilizing various semiconductors, particularly ones with extraordinarily high carrier mobility. These materials typically have small or even zero band gaps, which unavoidably causes significant leakage current or voltage loss in integrated circuits built with these semiconductors. To address this issue, C. Zhao et.al suggest and demonstrate a strengthened CMOS (SCMOS) logic style in this work using modified field-effect transistors (FETs)5. The goal is to achieve high performance by utilizing the high carrier mobility in these materials and to minimize the current leakage caused by their small band gap. By adding a second assistance gate close to the drain, it is possible to further reduce the potential barrier in the on-state and raise it in the off-state of conventional CMOS FETs. These modified asymmetric CMOS FETs, which exhibit perfect rail-to-rail output with negligible voltage loss, three orders of magnitude suppression of the static power consumption, and an operating speed comparable to or even higher than that of CMOS ICs, are used to build SCMOS integrated circuits (ICs). Although SCMOS is demonstrated here utilizing carbon nanotubes, this logic architecture can theoretically be implemented in integrated circuits (ICs) based on any small-band-gap semiconductor to offer great performance and low power consumption.
497: incomplete reference
Completed the reference as “de Sousa, M.S.M.; Liu, F.; Malard, M.; Qu, F.; Chen, W. Turning Graphene into Nodal-Line Semimetals by 496 Vacancy Engineering. APS March Meeting Abstracts, 2022, 1–7.”
512: remove "cite this"
removed
References:
- Chichinadze, D. V., Classen, L. & Chubukov, A. V. Nematic superconductivity in twisted bilayer graphene. Phys. Rev. B 101, 224513 (2020).
- Barbon, A., Tampieri, F., Barbon, A. & Tampieri, F. Identification of slow relaxing spin components by pulse EPR techniques in graphene-related materials. AIMS Mater. Sci. 2017 1147 4, 147–157 (2017).
- Alegaonkar, A., Alegaonkar, P. & Pardeshi, S. Exploring molecular and spin interactions of Tellurium adatom in reduced graphene oxide. Mater. Chem. Phys. 195, 82–87 (2017).
- Yan, W. et al. Long Spin Diffusion Length in Few-Layer Graphene Flakes. Phys. Rev. Lett. 117, 147201 (2016).
- Zhao, C. et al. Strengthened complementary metal-oxide-semiconductor logic for small-band-gap semiconductor-based high-performance and low-power application. ACS Nano 14, 15267–15275 (2020).

Reviewer 2 Report
Pawar, Duadi and Fixler set out to write a comprehensive review on carbon based spintronics. Spintronics is indeed one of the most important paradigm of the last 4 decades, with its most important technological maifestation, the magnetic hard drives. It also holds the promise of revolutionizing dynamic memory and computing, a promise that is long in the offing. What hinders the development of spintronics is materials related. In this sense, carbon based materials have long been considered as excellent candidates, starting with graphite intercalation compounds and polyacetylene, even before the coining of the term, Spintronics. A crucial physical property required for spintronics to work is a long and well understood spin relaxation time. Here, carbon based materials are considered to have a large advantage due to the reduced spin orbit coupling.
In this sense, the review sets out to discuss spin relaxation as one of its important stated objectives, but falls very much short. There is huge body of extremely relevant literature about spin relaxation in various carbon based materials that must be reviewed here, and in the current manuscript go completely unmentioned. Conduction electron spin resonance is an extremely fine tool to study spin relaxation in carbon based materials directly, indeed it is probably the only reliable tool.
It has detected hints of the spin density wave ground state through the antiferromagnetic resonance in the linear chain conducting polymers of fullerides doped 1:1 with alkali metals. Conduction electron spin resonance was used to determine sensitively the spin relaxation time in the superconducting state of various doped fullerides, with these studies leading to a unified theory of spin relaxation due to spin-orbit coupling. CESR was also used to determine the spin relaxation in single walled carbon nanotubes, potassium and other alkali doped carbon nanotubes, peapods, and led to determine the density of states in SWCNT and graphene. Furthermore, CESR was used to test the Elliott-Yafet spin relaxation in the graphite intercalation compunds KC8 as a model system for graphene. These studies culminated in the determination of the ultralong spin lifetime in light alkali doped graphene - the fundamental ingredient sought for spintronics.
I can not recommend publishing this review in its current state.
Author Response
We are thankful to the reviewers for sharing their valuable opinion on our work. The comments truly helped us to improve the content. We tried our level best to answer the comments of the reviewers through this revised form of the manuscript.
Reviewer: 2
Comments
Pawar, Duadi and Fixler set out to write a comprehensive review on carbon based spintronics. Spintronics is indeed one of the most important paradigm of the last 4 decades, with its most important technological maifestation, the magnetic hard drives. It also holds the promise of revolutionizing dynamic memory and computing, a promise that is long in the offing. What hinders the development of spintronics is materials related. In this sense, carbon based materials have long been considered as excellent candidates, starting with graphite intercalation compounds and polyacetylene, even before the coining of the term, Spintronics. A crucial physical property required for spintronics to work is a long and well understood spin relaxation time. Here, carbon based materials are considered to have a large advantage due to the reduced spin orbit coupling.
In this sense, the review sets out to discuss spin relaxation as one of its important stated objectives, but falls very much short. There is huge body of extremely relevant literature about spin relaxation in various carbon based materials that must be reviewed here, and in the current manuscript go completely unmentioned. Conduction electron spin resonance is an extremely fine tool to study spin relaxation in carbon based materials directly, indeed it is probably the only reliable tool.
It has detected hints of the spin density wave ground state through the antiferromagnetic resonance in the linear chain conducting polymers of fullerides doped 1:1 with alkali metals. Conduction electron spin resonance was used to determine sensitively the spin relaxation time in the superconducting state of various doped fullerides, with these studies leading to a unified theory of spin relaxation due to spin-orbit coupling. CESR was also used to determine the spin relaxation in single walled carbon nanotubes, potassium and other alkali doped carbon nanotubes, peapods, and led to determine the density of states in SWCNT and graphene. Furthermore, CESR was used to test the Elliott-Yafet spin relaxation in the graphite intercalation compunds KC8 as a model system for graphene. These studies culminated in the determination of the ultralong spin lifetime in light alkali doped graphene - the fundamental ingredient sought for spintronics.
Response: We thank the reviewer for the comment. We are happy to include the reference, and they have added them to the updated manuscript. We agree with the reviewer and added a new section 4 as “Spin relaxation in carbon-based materials” with the relevant references.

Reviewer 3 Report
The short review article titled “Recent advances in the spintronic application of carbon-based nanomaterials” is being reviewed. Although the content of this article is not deep enough, it can be used as a reference for beginners interested in related research issues. Therefore, the article is suggested to be published.
The comments are the following.
1. The authors have prepared the references for readers interested in the related research works.
2. Carbon-based nanomaterials should include organics. If the review article consists of an introduction to organic spintronics will be much better. Otherwise, the title should be changed.
Author Response
We are thankful to the reviewers for sharing their valuable opinion on our work. The comments truly helped us to improve the content. We tried our level best to answer the comments of the reviewers through this revised form of the manuscript.
Reviewer 3:
Comments
The short review article titled "Recent advances in the spintronic application of carbon-based nanomaterials" is being reviewed. Although the content of this article is not deep enough, it can be used as a reference for beginners interested in related research issues. Therefore, the article is suggested to be published.
The comments are the following.
- The authors have prepared the references for readers interested in the related research works.
2. Carbon-based nanomaterials should include organics. If the review article consists of an introduction to organic spintronics will be much better. Otherwise, the title should be changed.
Response: We thank the reviewer for the comment. We are happy to include the suggestion. We agree with the reviewer and added a new section 5 as “Organic spintronics” with the relevant references.

Round 2
Reviewer 2 Report
The authors have expanded their review in several parts. In particular they attempted to include a section on the spin-relaxation/spin-lifetime in carbon based materials, however this section feels rather wikipedia-like, it contains several unfinished sentences, and includes very limited and somewhat irrelevant references. As I indicated in my previous report, there is a large body of CESR studies of fullerides, SWNTs and alkali doped graphene which ought to be addressed.
Author Response
We are thankful to the reviewer for sharing his/her valuable opinion on our work. The comments truly helped us to improve the content. We tried our level best to answer the comments of the reviewer through this revised form of the manuscript.
Reviewer 2:
Comments
The authors have expanded their review in several parts. In particular they attempted to include a section on the spin-relaxation/spin-lifetime in carbon based materials, however this section feels rather wikipedia-like, it contains several unfinished sentences, and includes very limited and somewhat irrelevant references. As I indicated in my previous report, there is a large body of CESR studies of fullerides, SWNTs and alkali doped graphene which ought to be addressed.
Response: We thank the reviewer for the comment. Again, we agree with the reviewer and we rewrite this section. However, as a review paper we cannot see a problem with wikipedia-like section, hope to get the same citations as Wikipedia… We are happy to include the reference, and have added them to the updated manuscript. We agree with the reviewer and updated section 4 “Spin relaxation in carbon-based materials” with the relevant references on CESR studies of fullerides, SWNTs and alkali doped graphene.
We would like to ask the reviwer to reconsider his low rating for the significant contribution of our manuscript to the field.

Round 3
Reviewer 2 Report
The authors have expanded their review considerably, making the review a useful addition to the field.